# Global Explainability of GNNs via Logic Combination of Learned Concepts

**Steve Azzolin**
University of Trento, Trento
`steve.azzolin@studenti.unitn.it`

**Antonio Longa**
University of Trento, Trento
`antonio.longa@unitn.it`

**Pietro Barbiero**
University of Cambridge, Cambridge
`pb737@cam.ac.uk`

**Pietro Liò**
University of Cambridge, Cambridge
`pl219@cam.ac.uk`

**Andrea Passerini**
University of Trento, Trento
`andrea.passerini@unitn.it`

## Abstract

While instance-level explanation of GNN is a well-studied problem with plenty of approaches being developed, providing a *global* explanation for the behaviour of a GNN is much less explored, despite its potential in interpretability and debugging. Existing solutions either simply list local explanations for a given class, or generate a synthetic prototypical graph with maximal score for a given class, completely missing any combinatorial aspect that the GNN could have learned. In this work we propose GLGExplainer (Global Logic-based GNN Explainer), the first Global Explainer capable of generating explanations as arbitrary Boolean combinations of learned graphical concepts. GLGExplainer is a fully differentiable architecture that takes local explanations as inputs and combines them into a logic formula over graphical concepts, represented as clusters of local explanations. Contrary to existing solutions, GLGExplainer manages to provide accurate and human-interpretable global explanations in both synthetic and real world datasets.

## 1 Introduction

Graph Neural Networks (GNNs) have become increasingly popular for predictive tasks on graph structured data. However, as many other deep learning models, their inner working remains a black box. The ability to understand the reason for a certain prediction represents a critical requirement for any decision-critical application, thus representing a big issue for the transition of such algorithms from benchmarks to real-world critical applications.

Over the last years, many works proposed Local Explainers [1–9] to explain the decision process of a GNN in terms of factual explanations often represented as subgraphs for each sample in the dataset. Overall, they shed light over *why* the network predicted a certain value for a specific input sample. However, they still lack a global understanding of the model. Global Explainers, on the other hand, are aimed at capturing the behaviour of the model as a whole, abstracting individual noisy local explanations in favor of a single robust overview of the model. However, despite their potential in interpretability and debugging little has been done in this direction [10]. GLocalX [11] is a general solution to produce global explanations of black-box models by hierarchically aggregating local explanations into global rules via an heuristic-based iterative procedure. This solution is however not readily applicable to GNNs as it requires local explanations to be expressed as logical rules. Yuan et al. [10] proposed to frame the Global Explanation problem for GNN as a form of input optimization,

Azzolin, S. et al., Global Explainability of GNNs via Logic Combination of Learned Concepts (Extended Abstract). Presented at the First Learning on Graphs Conference (LoG 2022), Virtual Event, December 9–12, 2022.

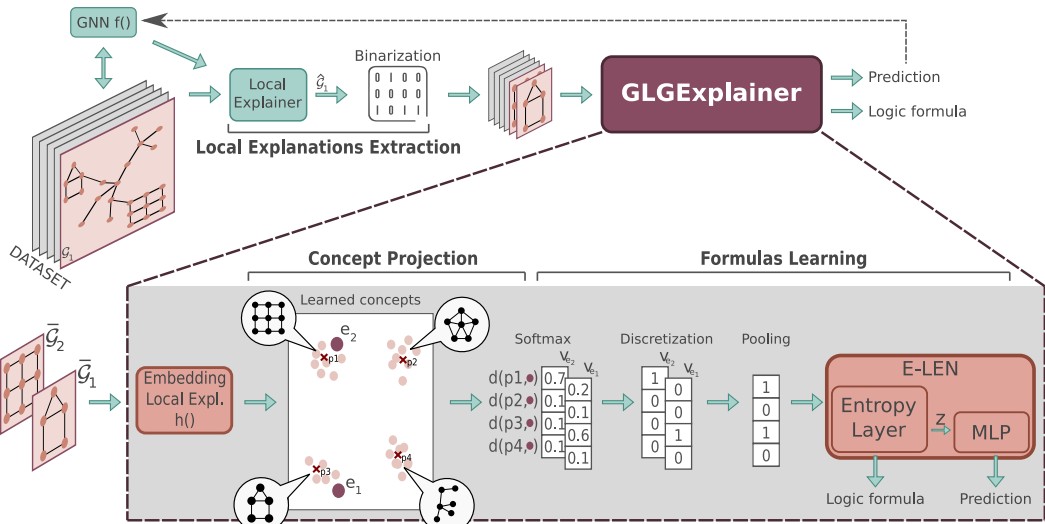

**Figure 1:** Illustration of the proposed method for a task of binary classification. Each step is described in detail in Section 2

.

similarly as done for some vision models [12], using policy gradient to generate synthetic prototypical graphs for each class. The approach requires prior domain knowledge, which is not always available, to drive the generation of valid prototypes. Additionally, it cannot identify any compositionality in the returned explanation, and has no principled way to generate alternative explanations for a given class.

Concept-based Explainability [13–15] is a parallel line of research where explanations are constructed using "concepts" i.e., intermediate, high-level and semantically meaningful units of information commonly used by humans to explain their decisions. Concept Bottleneck Models [16] and Prototypical Part networks [17] are two popular architectures that leverage concept learning to learn explainable-by-design neural networks. Both approaches have been recently adapted to GNNs [18, 19]. However, these solutions are not conceived for explaining already learned GNNs.

**Our contribution** consists in the first Global Explainer for GNNs which *i)* provides a Global Explanation in terms of logic formulas, extracted by combining in a fully differentiable manner graphical concepts derived from local explanations; *ii)* is faithful to the data domain, i.e., the logic formulas, being derived from local explanations, are intrinsically part of the input domain without requiring any prior knowledge. We validated our approach on both synthetic and real-world datasets, showing that our method is able to accurately summarize the behaviour of the model to explain, while providing explanations in terms of concise logic formulas.

## 2 Proposed Method

Our proposed Global Explainer, named GLGExplainer (Global Logic-based GNN Explainer), is summarized in Figure 1. In the following we will describe each step in greater detail.

**Local Explanations Extraction:** The first step of our pipeline consists in extracting local explanations. Let $\text{LEXP}(f, \mathcal{G}) = \hat{\mathcal{G}}$ be the weighted graph obtained by applying the local explainer $\text{LEXP}$ to generate a local explanation for the prediction of the GNN $f$ over the input graph $\mathcal{G}$. In principle, every Local Explainer whose output can be mapped to a subgraph of the input sample is compatible with our pipeline [1–6]. Nonetheless, in this work, we relied on PGExplainer [2] since it allows the extraction of arbitrary disconnected motifs as explanations and it gave excellent results in our experiments. By binarizing the output of the local explainer $\hat{\mathcal{G}}$ with threshold $\theta \in \mathbb{R}$ we achieve a set of connected components $\bar{\mathcal{G}}_i$ such that $\bigcup_i \bar{\mathcal{G}}_i \subseteq \hat{\mathcal{G}}$. For convenience, we will henceforth refer to each of these $\bar{\mathcal{G}}_i$ as local explanation. Given that we want to emulate the behaviour of $f$ on correctly predicted samples, we will discard every input graph $\mathcal{G}$ belonging to wrongly predicted samples.

**Table 1:** Mean and standard deviation for Fidelity, Formula Accuracy and Concept Purity computed on the Test set over 5 runs with different random seeds. Since Concept Purity is computed for every cluster independently, here we report mean and standard deviation for the best run only.

| Dataset | Fidelity | Formula Accuracy | Concept Purity |
|---------|----------|------------------|----------------|
| BAMultiShapes | 0.99 $\pm$0.00 | 0.99 $\pm$0.00 | 0.85 $\pm$0.22 |
| Mutagenicity | 0.85 $\pm$0.01 | 0.85 $\pm$0.01 | 0.99 $\pm$0.01 |

The result of this extraction thus consists in a list $D$ of local explanations. More details about the binarization are available in the Appendix.

**Embedding Local Explanations:** The following step consists in learning an embedding for each local explanation that allows to cluster together functionally similar local explanations. This can be achieved with a standard GNN $h$ which maps any graph $\bar{\mathcal{G}}$ into a fixed-sized embedding $h(\bar{\mathcal{G}}) \in \mathbb{R}^d$. Since each local explanation $\bar{\mathcal{G}}$ is a subgraph of an input graph $\mathcal{G}$, in our experiments we used the original node features of the dataset. The outcome of this aggregation consists in a set $E = \{h(\bar{\mathcal{G}}), \ \forall \bar{\mathcal{G}} \in D\}$ of graph embeddings.

**Concept Projection:** Inspired by previous works on prototype learning [20, 21], we project each graph embedding $e \in E$ into a set $P$ of $m \in \mathbb{N}$ prototypes $\{p_i \in \mathbb{R}^d | i = 1, \ldots, m\}$ via a distance function $d(p_i, e) = softmax \left( log(\frac{\|e-p_1\|^2+1}{\|e-p_1\|^2+\epsilon}), \ldots, log(\frac{\|e-p_m\|^2+1}{\|e-p_1\|^2+\epsilon}) \right)_i$. Prototypes are initialized randomly from a uniform distribution and are learned along with the other parameters of the architecture. As training progresses, the prototypes will align as prototypical representations of every cluster of local explanations, which will represent the final groups of graphical concepts. The output of this projection is thus a set $V = \{v_e, \ \forall e \in E\}$ where $v_e = [d(p_1, e), .., d(p_m, e)]$ is a vector containing the normalized probabilities of local explanation $i$ belonging to the $m$ concepts, and will be henceforth referred to as *concept vector*.

**Formulas Learning:** The final step consists of an E-LEN, i.e., a Logic Explainable Network [22] implemented with an Entropy Layer as first layer [23]. An E-LEN learns to map a concept activation vector to a class while encouraging a sparse use of concepts that allows to reliably extract Boolean formulas emulating the network behaviour. We train an E-LEN to emulate the behaviour of the GNN $f$ feeding it with the graphical concepts extracted from the local explanations. Given a set of local explanations $\bar{\mathcal{G}}_a \ldots \bar{\mathcal{G}}_{n_i}$ for an input graph $\mathcal{G}_i$ and a corresponding set of the concept vectors $v_a \ldots v_{n_i}$, we aggregate the concept vectors via a pooling operator and feed the resulting aggregated concept vector to the E-LEN, providing $f(\mathcal{G}_i)$ as supervision. In our experiments we used a max-pooling operator. Thus, the Entropy Layer learns a mapping from the pooled concept vector to (i) the embeddings $z$ (as any linear layer) which will be used by the successive MLP for matching the predictions of $f$. (ii) a truth table $T$ explaining how the network leveraged concepts to make predictions for the target class. Since the input pooled concept vector will constitute the premise in the truth table $T$, a desirable property to improve human readability is discreteness, which we achieved using the Straight-Through (ST) trick used for discrete Gumbel-Softmax Estimator [24]. In practice, we compute the forward pass discretizing each $v_i$ via *argmax*, then, in the backward pass to favor the flow of informative gradient we use its continuous version.

**Supervision Losses:** Our proposed GLGExplainer is trained end-to-end with the following loss: $L = L_{surr} + \lambda_1 L_{R1} + \lambda_2 L_{R2}$, where $L_{surr}$ corresponds to a Focal BCELoss [25] between the prediction of our E-LEN and the predictions to explain, while $L_{R1}$ and $L_{R2}$ are respectively aimed to push every prototype to be close to at least one local explanation and to push each local explanation to be close to at least one prototype [20]. The losses are defined as follows:

$$L_{surr} = -y(1-p)^\gamma \log p - (1-y)p^\gamma \log(1-p) \tag{1}$$

$$L_{R1} = \frac{1}{m} \sum_{j=1}^{m} \min_{\bar{\mathcal{G}} \in D} \|p_j - h(\bar{\mathcal{G}})\|^2 \tag{2}$$

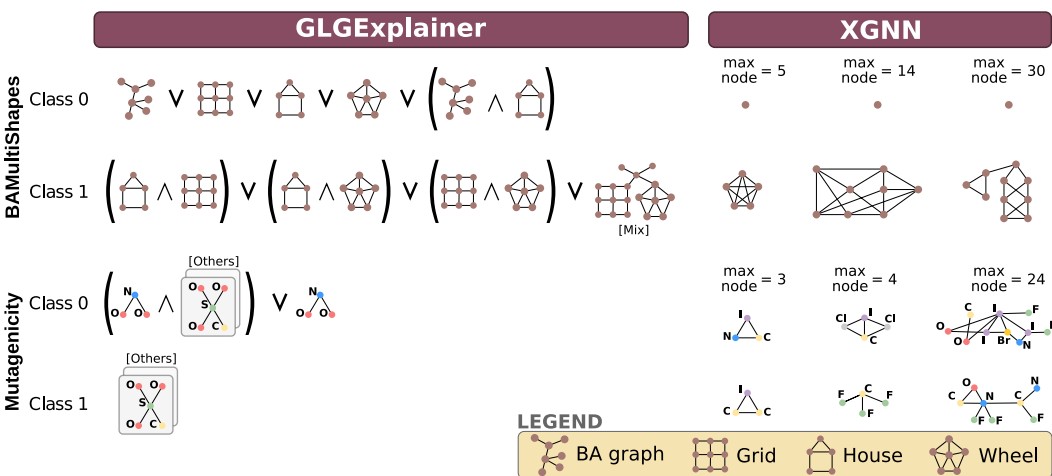

**Figure 2:** Global explanations of GLGExplainer (ours) and XGNN. The class probability predicted by XGNN for the generated explanations is around 1 for every explanation, except for Class 0 of BAMultiShapes where it was not able to generate a graph with confidence $\geq 0.5$.

$$L_{R2} = \frac{1}{|D|} \sum_{\bar{\mathcal{G}} \in D} \min_{j \in [1,m]} \|p_j - h(\bar{\mathcal{G}})\|^2 \tag{3}$$

where $p$ and $\gamma$ represent respectively the probability for positive class prediction and the *focusing* parameter which controls how much to penalize hard examples.

## 3 Experiments

We tested our proposed approach on two datasets, namely:

**BAMultiShapes:** BAMultiShapes is a newly introduced extension of some popular synthetic benchmarks [1] aimed to assess the ability of a Global Explainer to deal with logical combinations of concepts. In particular, we created a dataset composed of Barabási-Albert (BA) graphs with attached in random positions the following network motifs: house, grid, wheel. Class 0 contains plain BA graphs and BA graphs enriched with a house, a grid, a wheel, or the three motifs together. Class 1 contains BA graphs enriched with a house and a grid, a house and a wheel or a wheel and a grid.

**Mutagenicity:** The Mutagenicity dataset is a collection of molecule graphs where each graph is labelled as either having a mutagenic effect or not. Based on [26], the mutagenicity of a molecule is correlated with the presence of electron-attracting elements conjugated with nitro groups (e.g. *NO2*).

For Mutagenicity we replicated the model accuracy and the local explanations presented in [2], while for BAMultiShapes we trained until convergence a 3-layers GCN. Details about the implementation and the pre-processing of local explanations, along with model accuracies, are in the Appendix.

In order to show the robustness of our proposed methodology, we have evaluated GLGExplainer on a number of metrics, namely: FIDELITY, FORMULA ACCURACY, and CONCEPT PURITY. A detailed description of those metrics can be found in the Appendix. Table 1 reports the results in terms of the three metrics, showing how GLGExplainer manages to provide reliable explanations under all these perspectives. Note that XGNN [10], the only available competitor for global explanations of GNN, cannot be evaluated according to these metrics. Figure 2 presents the final global explanations where we substituted each literal with its corresponding prototypical graphical concept, and report the explanations generated by XGNN for comparison. It's easy to see that GLGExplainer produces highly interpretable explanations that match the ground-truth formula (for BAMultiShapes) and existing knowledge (for Mutagenesis) with remarkable accuracy. It is worth mentioning that the global explanations for Class 0 of BAMultiShapes do not comprise the case with all three motifs together. We observed that the reason resides in the GNN to explain failing at classifying every sample with such structure. So, GLGExplainer is effectively explaining the GNN $f$ and not simply

the dataset structure. Conversely, XGNN fails to generate interpretable explanations in most cases. Details about concepts compositions and formula extraction are available in the Appendix.

## 4    Discussion & Conclusions

Given the results presented in the section above, it is worth noting that concept clusters emerge solely based on the supervision defined in Section 2, while no specific supervision was added to cluster local explanations based on their similarity. Further details about the clusters' composition are available in the Appendix. Overall, the results confirm the ability of GLGExplainer in providing logic formulas, expressed over learned graphical concepts, which are accurately summarizing the global behaviour of the model, whereas the existing XGNN fails at providing concise and faithful explanations.

## Acknowledgements

This research was partially supported by TAILOR, a project funded by EU Horizon 2020 research and innovation programme under GA No 952215

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

# A Appendix

## A.1 Training the GNN $f$

For both BAMultiShapes and Mutagenicity we relied on the codebase provided by [2] for training the GNN $f$ to explain and to train the Local Explainer. For BAMultiShapes we trained a 3-layers GCN [27] (20-20-20 hidden units) with mean graph pooling for the final prediction, whereas for Mutagenicty we reproduced the results of [2]. A summary of model's performance is available in Table 2. Despite the high accuracy over BAMultiShapes, after a closer look we observed that the network did not actually learn the *All* concept, i.e., the three motifs together. Such detailed view is available in Table 3. This explains why the global explanations in Figure 2 Class 0 do not present such concept.

**Table 2:** GNN accuracies for BAMultiShapes and Mutagenicity. The results for Mutagenicity are in line with the one reported in [2].

| Split | BAMultiShapes | Mutagenicity |
|-------|---------------|--------------|
| Train | 0.94 | 0.87 |
| Val | 0.94 | 0.86 |
| Test | 0.99 | 0.86 |

**Table 3:** Accuracy of the model on the train set of BAMultiShapes with respect to every combination of motifs to be added to the Barabási-Albert base graph. *H, G, W* stand respectively for House, Grid, and Wheel.

| Motifs | | | Class 0 | | | | Class 1 | |
|--------|-----------|-----|-----|-----|-------|-----------|-----------|-----------|
| | $\emptyset$ | $H$ | $G$ | $W$ | $All$ | $H+G$ | $H+W$ | $G+W$ |
| **Accuracy** (%) | 1.0 | 1.0 | 0.85 | 1.0 | 0.0 | 1.0 | 0.98 | 1.0 |

## A.2 Local Explanations Processing

As detailed in [2], the output of PGExplainer consists in a weighted edge mask $w_{ij} \in \mathcal{V} \times \mathcal{V}$ where each $w_{ij}$ is the likelihood of the edge being an important edge. For Mutagenicity, we sticked to the original implementation which was correctly able to reproduce the results presented in the paper [2]. The only difference resides in the procedure for cutting the explanation, which is needed to remove from the final local explanation the edges which were assigned low scores. The authors in [2] limited their analysis to graphs that were containing the ground truth motifs, and proposed to just keep the top-k edges. We, instead, selected the numeric threshold $\theta \in \mathbb{R}$ which maximises the F1 score of the explainer over all graphs. Afterwards, such threshold will be used to cut out the irrelevant edges, by applying the indicator function $\mathbf{1}_{w_{ij} \geq \theta}$ to the edge mask. The resulting edge mask is thus the binary adjacency matrix of the final explanation. For BAMultiShapes, however, we adopted a dynamic algorithm to select $\theta$ that does not require any prior knowledge about the ground truth motifs. This algorithm resembles the elbow-method, i.e., for each local explanation choose as $\theta$ the first value that is different enough from the previous ordered values. Figure 3 shows some examples for each dataset along with their local explanations in bold.

## A.3 The GLGExplainer

The reference implementation of our Local Explanation Embedder $h$ is constituted by a 2-layers GIN [28] network with 20 hidden units, followed by a non-linear combination of max, mean, and

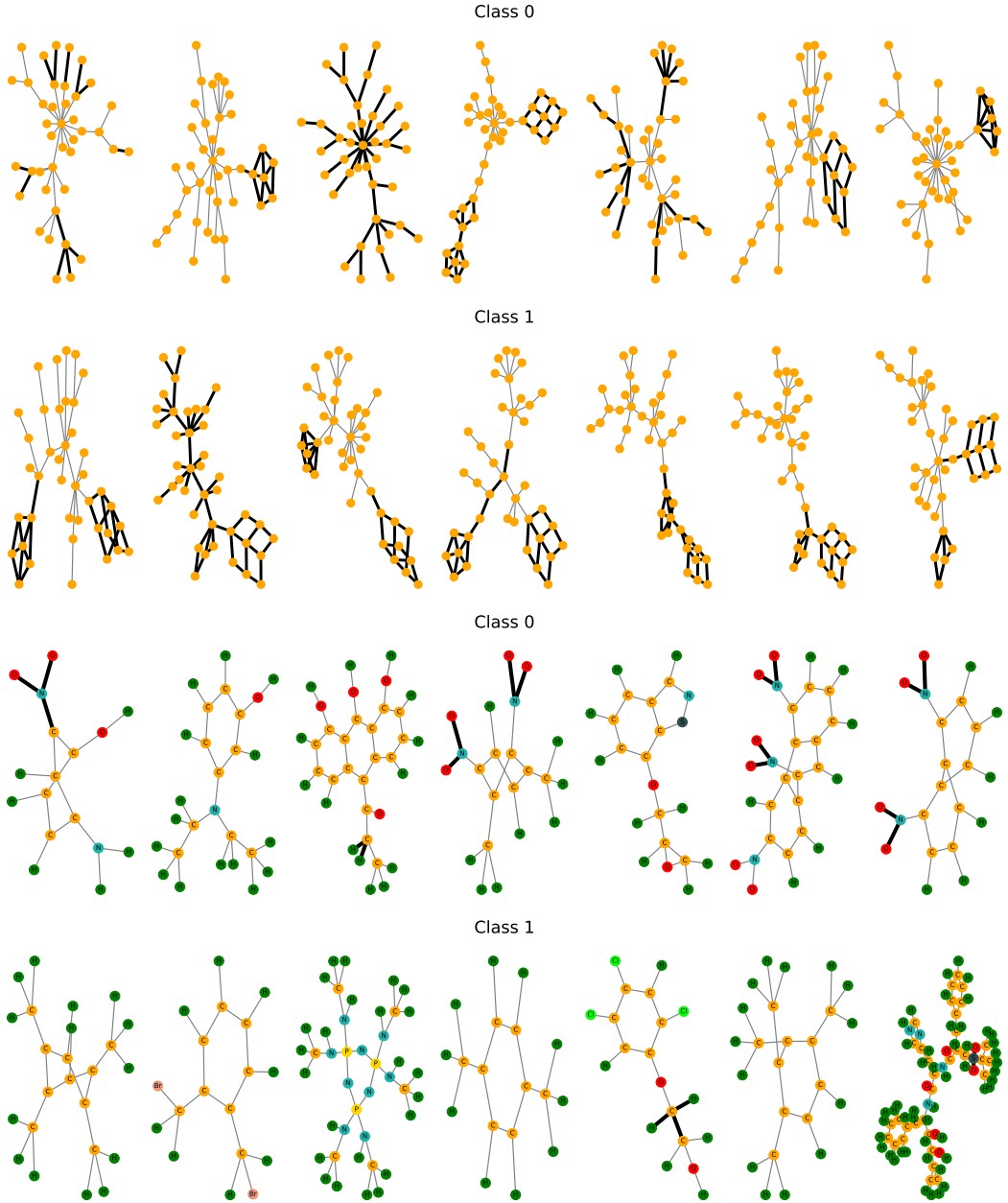

**Figure 3:** Random examples of input graphs along with their explanations in bold as extracted by PGExplainer, for respectively BAMultiShapes and Mutagenicity.

sum graph pooling. We chose a number $m$ of 6 and 2 prototypes for, respectively, BAMultiShapes and Mutagenicity, keeping the dimensionality $d$ to 10. We trained using ADAM optimizer with early stopping and with a learning rate for $h$ and the prototypes $P$ of $1e^{-3}$, while for the E-LEN of $5e^{-4}$. The batch size is set to 128, while the auxiliary loss coefficients $\lambda_1$ and $\lambda_2$ are chosen via cross-validation and set respectively to 0.09 and 0.00099, while the focusing parameter $\gamma$ is kept fixed at 2. The E-LEN is constituted by the input Entropy Layer ($Entr.Layer : \mathbb{R}^m \to \mathbb{R}^{10}$), a hidden layer ($HiddenLayer : \mathbb{R}^{10} \to \mathbb{R}^5$), and the output layer with LeakyReLU activation function.

In the rest of this section we provide an ablation study to demonstrate the effectiveness of the Focal loss, the Discretization trick, and the impact of the number of prototypes in use.

**Focal loss**: Figure 4 presents a comparison of the learning curve for BAMultiShapes showing that using Focal loss with a *focusing* parameter of 2 helps to achieve a faster convergence while not being detrimental for the overall performances.

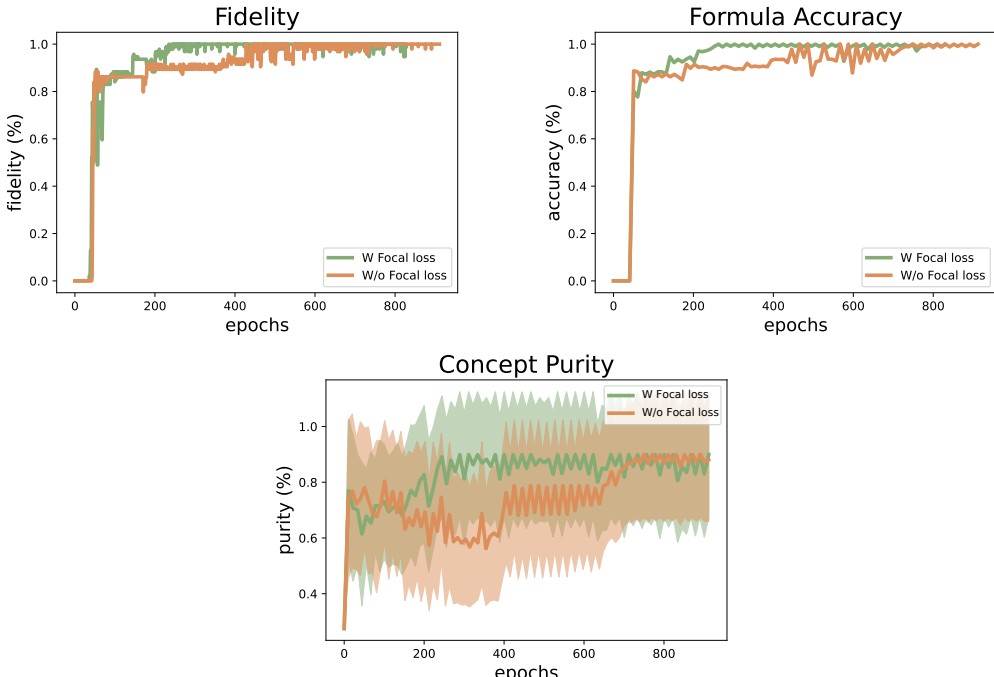

**Figure 4:** Learning curves for BAMultiShapes with and without Focal loss. Results show that the Focal loss, with a focusing parameter set to 2, helps to achieve faster convergence while not reducing the final performances.

**Number of prototypes**: An effective approach to select an appropriate value $m$ for the number of prototypes in use is via cross-validation, and by selecting the smallest $m$ which achieves a competitive fidelity. In Figure 5 we show how different values of $m$ impact the Fidelity and the Formula Accuracy.

**Discretization trick**: The Discretization trick was introduced in Section 2 to enforce a discrete prototype assignment, something essential for an unambiguous definition of the concepts on which the formulas are based on. In Figure 6 we show for BAMultiShapes that this trick is also effective in improving the overall performance of GLGExplainer, since it forces the hidden layers of the E-LEN to just exploit the information relative to the closest prototype, while not relying on other positional information. Thus, the E-LEN's predictions are much more aligned with the discrete formulas being extracted. In the Figure we further compare against a plain model without Discretization and against the addition to the overall loss of an entropy loss over the concept vector (Concept Entropy loss) with different scaling parameters $\lambda_3 \in \{0.01, 0.1\}$. This Concept Entropy loss (CE loss) pushes the pre-pooling concept vector to have low entropy, thus effectively pushing every local explanation to be assigned with confidence to just one prototype.

### A.4  Cluster Composition & Formulas Renaming

To effectively explore the content of each local explanations cluster, we plot in Figure 7 some random elements for each dataset. In most cases, the clusters contain atomic motifs (House, Grid, NO2, etc..) while in others the embedder $h$ clustered together heterogeneous motifs. This is particularly evident for the cluster relative to the prototype $p_3$ of BAMultiShapes in which every local explanation comprising two atomic motifs are aggregated. The reason for this behaviour is that we are aggregating local explanation solely based on the ability of the E-LEN to emulate the predictions of $f$. Thus, since the simultaneous presence of two motifs appears only in Class 1, one single cluster aggregating all these *mixed* local explanations is enough for maximizing the performances. This is also the reason

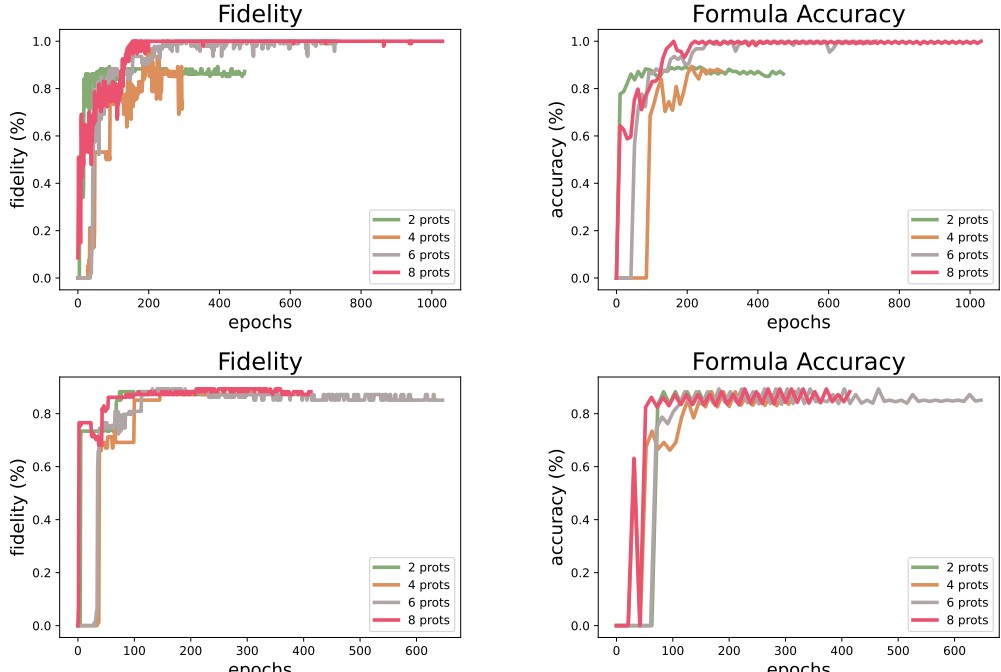

**Figure 5:** Ablation study on the number of prototypes to use. The first row is referred to BAMulti-Shapes, while the second to Mutagenicity.

**Table 4:** Raw formulas as extracted by the Entropy Layer. Each formula was rewritten following the Closed-World Assumption for convenience.

| Dataset | Raw Formulas |
|---------|--------------|
| BAMultiShapes | $\text{Class}_0 \iff P_0 \vee P_5 \vee P_1 \vee P_4 \vee P_2 \vee (P_4 \wedge P_2)$ 
 $\text{Class}_1 \iff P_3 \vee (P_5 \wedge P_2) \vee (P_5 \wedge P_1) \vee (P_2 \wedge P_1)$ |
| Mutagenicity | $\text{Class}_0 \iff P_1 \vee (P_0 \wedge P_1)$ 
 $\text{Class}_1 \iff P_0$ |

for the high variability in Concept Purity reported in Table 1, since it is computed considering the Purity in terms of labelled atomic motifs. For completeness, we additionally report in Figure 8 a 2D PCA-reduced view of the embedding space, annotated with the prototypes position.

Given that the default implementation of the Entropy layer returns formulas expressed in terms of the single concepts in input, Figure 7 is also useful to rename each literal into its corresponding graphical concept. Table 4 shows an example of such raw formulas, while Figure 2 presents the final formulas after replacing each raw name with the corresponding graphical concept.

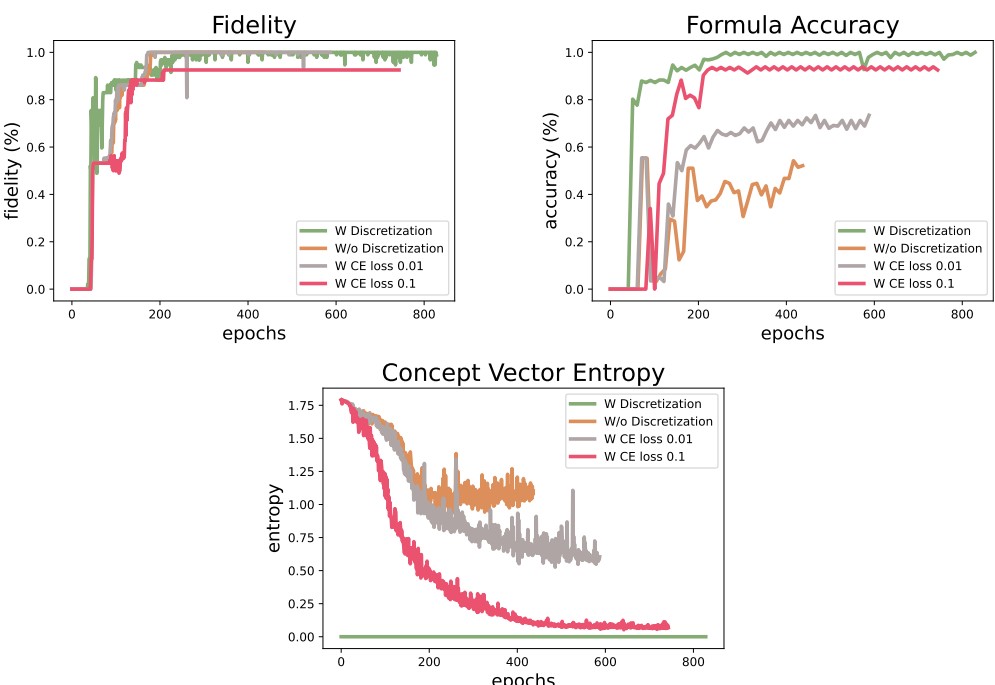

**Figure 6:** Ablation study for the impact of the Discretization trick, discussed in Section 2. We compared the performances with and without it, and against the addition of a Concept Entropy loss (CE loss) with different scaling parameters $\lambda_3$.

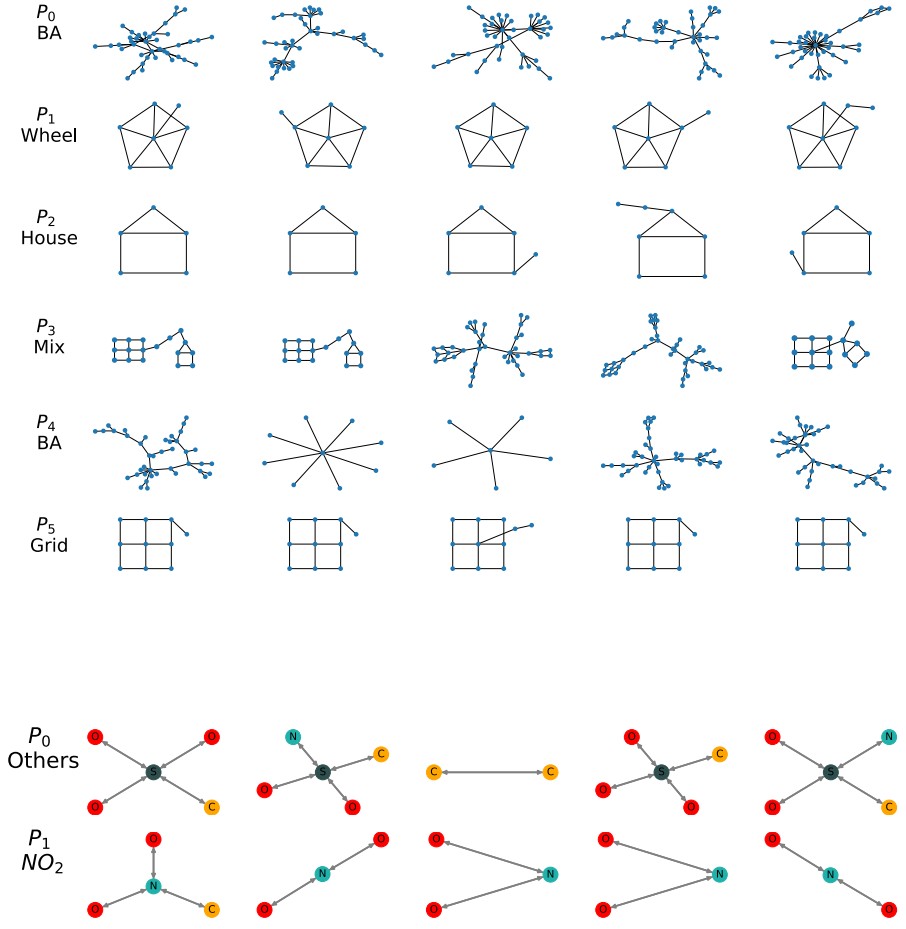

**Figure 7:** Random representative elements for each prototype in BAMultiShapes and Mutagenicity.

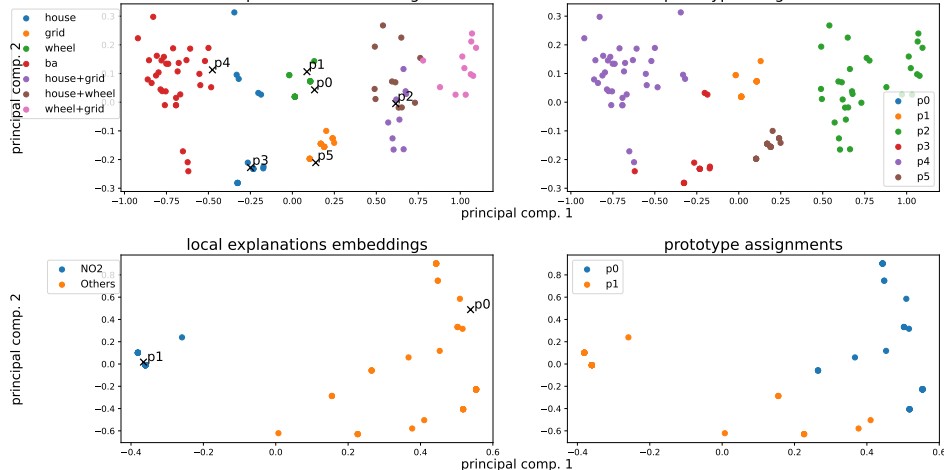

**Figure 8:** 2D view of the embedding space annotated with prototypes positions. The first line is referred to BAMultiShapes, while the second to Mutagenicity.

### A.5 Benefits and Limitations

As previously discussed, the proposed GLGExplainer is inherently faithful to the data domain since it processes local explanations provided by a Local Explainer. However, the quality of those local explanations, in terms of representativeness and discriminability with respect to the task-specific class, has a direct effect on the Fidelity. If the generated concept vector does not exhibit any class-specific pattern, then the E-LEN will not be able to emulate the predictions of the model to explain. Despite being a potential limitation of GLGExplainer, this can actually open to the possibility of using the Formula Accuracy as a proxy of local explanations quality, which is notoriously difficult to assess. We leave this investigation to future work. Despite tailoring our discussion on graph classification, our approach can be readily extended to any kind of classification task on graphs, provided that a suitable Local Explainer is available.

### A.6 Evaluation Metrics

Here we will describe in mode detail the metrics briefly introduced in Section 3:

- Fidelity measures the accuracy between the prediction of the E-LEN and the one of the GNN to explain. It is computed as the accuracy between the class predictions of the E-LEN and the GNN $f$.

- Formula Accuracy represents how well the learned formulas can correctly predict the class labels. To compute this metric, we treat the final formulas as a classifier that given an input concept vector predicts the class corresponding to the clause evaluated to true. In the cases in which either no clause or more clauses of different classes are evaluated to be true, the sample is always considered as wrongly predicted.

- Concept Purity is computed for every cluster independently and measures how good the embedding is at clustering the local explanations. Specifically, it requires each local explanation to be annotated with a label, which in our cases corresponds to the typology of the motif represented by the local explanation. Then, the computation of the metric can be summarized by:

$$ConceptPurity(C_i) = \frac{count\_most\_frequent\_label(C_i)}{|C_i|} \tag{4}$$

where $C_i$ corresponds to the cluster having $p_i$ as prototype (i.e., the cluster containing every local explanation associated to prototype $p_i$ by the distance function $d(.,.)$ described in Section 2). $count\_most\_frequent\_label(C_i)$ instead returns the number of local explanations annotated with the most present label in cluster $C_i$. The Concept Purity results reported in Table 1 are computed by taking the mean and the standard deviation across the $m$ clusters.

### A.7 Additional Experiments

In this section, we report further experimental results in addition to those presented in Section 3.

#### A.7.1 Multi-Class Dataset

To challenge GLGExplainer also in a multi-class setting, we extended the previously introduced BAMultiShapes dataset with an additional class, constituted by the usual random BA base graph with attached house-like motifs and a cycle of length 6. We will henceforth refer to this dataset as *BAMultiShapesMC*. As for BAMultiShapes, node features are represented by a fixed vector with values 0.1. Similarly as done for BAMultiShapes, we trained an extension of the 3-layers GCN used before, using both a max and mean aggregator for graph pooling. The resulting accuracies are $0.95\%$ and $0.98\%$ for respectively the train and test set. We kept the same setting for extracting and processing local explanations, with the only difference that we used 128 hidden units for PGExplainer, instead of the original 64, to favour the extraction of good local explanations which otherwise were of very low quality. For GLGExplainer, we kept almost all of the hyper-parameters presented in Appendix A.3. We found, however, to be very beneficial for the final embedding to use $m = 8$. Note that this is a reasonable modification, given that we added completely new motifs and the overall dataset composition is more complex.

In Table 5 we report the results over 5 different random seeds, while in Figure 10 we provide an illustration of the PCA 2D-reduced embedding for the best run along with the relative formulas

**Table 5:** Mean and standard deviation for Fidelity, Formula Accuracy and Concept Purity computed on the Test set over 5 runs with different random seeds. Since the Concept Purity is computed for every cluster independently, here we report mean and standard deviation for the best run only.

| **Dataset** | Fidelity | Formula Accuracy | Concept Purity |
|---|---|---|---|
| BAMultiShapesMC | $0.97 \pm 0.02$ | $0.97 \pm 0.01$ | $0.82 \pm 0.23$ |

**Table 6:** Raw formulas as extracted by the Entropy Layer along with their test Fidelity.

| **Dataset** | **Raw Formulas** | | **Fidelity** |
|---|---|---|---|
| BAMultiShapesMC | $Class_0 \iff$ | $P_0 \vee P_2 \vee P_3 \vee P_5 \vee P_7$ | 0.98 |
| | $Class_1 \iff$ | $P_1 \vee P_6 \vee (P_0 \wedge P_2) \vee (P_0 \wedge P_5) \vee$ $(P_0 \wedge P_7) \vee (P_5 \wedge P_7) \vee (P_6 \wedge P_7) \vee$ $(P_5 \wedge P_6) \vee (P_0 \wedge P_4) \vee (P_0 \wedge P_6) \vee$ $(P_2 \wedge P_6) \vee (P_2 \wedge P_7) \vee (P_3 \wedge P_5 \wedge P_6)$ | |
| | $Class_2 \iff$ | $P_4 \vee (P_3 \wedge P_5) \vee (P_2 \wedge P_3) \vee$ $(P_3 \wedge P_4) \vee (P_2 \wedge P_3 \wedge P_5)$ | |

in Table 6. As it is possible to inspect from the output of the E-LEN, the resulting extracted raw formulas are still well representing the underlying ground truth modelling correctly the presence of the new class, despite containing some additional noise. For example, given the overlapping between some prototype assignments (like the cluster of $P_5$ that, even if representing the vast majority of BA base graphs, it contains some spurious houses, or similarly for $P_3$ that despite containing every local explanation representing a circle, contains a few BA graphs) the E-LEN learned some spurious clauses which are not correctly modelling the underlying ground truth. Those cases represent however the real minority of cases, being the overall formulas and Fidelity well aligned with the results obtained for BAMultiShapes, where the quality of local explanations allowed more distinctive clusters. Note indeed that as described in [23] and implemented in the official codebase[1], it is possible to rank the clauses in the truth table $T$ created by the E-LEN by their support (for how many samples they hold), allowing to select only the top-ranked clauses either by evaluating on a validation set, or by specifying a minimum support. Nonetheless, we did not apply such filter in order to stick to the experimental setting previously defined. In the same vein, despite the possibility of arbitrarily augmenting the local explanations' node features with any hand-crafted feature in order to better separate the clusters, we kept only the original datasets' node features.

As mentioned in Appendix A.5, GLGExplainer can be used to get insights into the Local Explainer in use. To this end, it is possible to understand the reason behind, for example, the clause $P_1$ for $Class_1$ in Table 6 by the fact that each of the $\sim 20$ samples in this cluster corresponds to the external border of the grid-like motif, which are extracted for the vast majority in $Class_1$. This means that, in this specific case, PGExplainer had a bias in extracting solely the border of the grid motif in $Class_1$, ignoring the other motif present in the same sample (recall that every sample of $Class_1$ has two motifs). An example of such motif along with the overall sample-wise local explanations are reported in Figure 9.

### A.7.2 Impact of the GNN Architecture on the Embedder $h$

In our previous experiments we implemented the Local Explanations Embedder $h$ as a 2-layers GIN network. However, any compatible GNN architecture can be used instead. In Figure 11 we present an ablation study testing different GNN architectures, and comparing their respective performances in terms of Fidelity and Concept Purity. For every architecture under analysis we kept the number of layers equal to 2. Specifically, we tested GCN [27], GIN [28], SAGE [29], and GAT [30]. For BAMultiShapes, in which are present different and heterogeneous motifs, architectures with a sum local aggregator like SAGE and GIN seem to be preferable over GCN, where presumably the node-degree normalization in its propagation rule limits the richness of the learned representation. Given the absence of informative node features, GAT is not able to properly learn a useful embedding. For

---

[1]https://pypi.org/project/torch-explain/

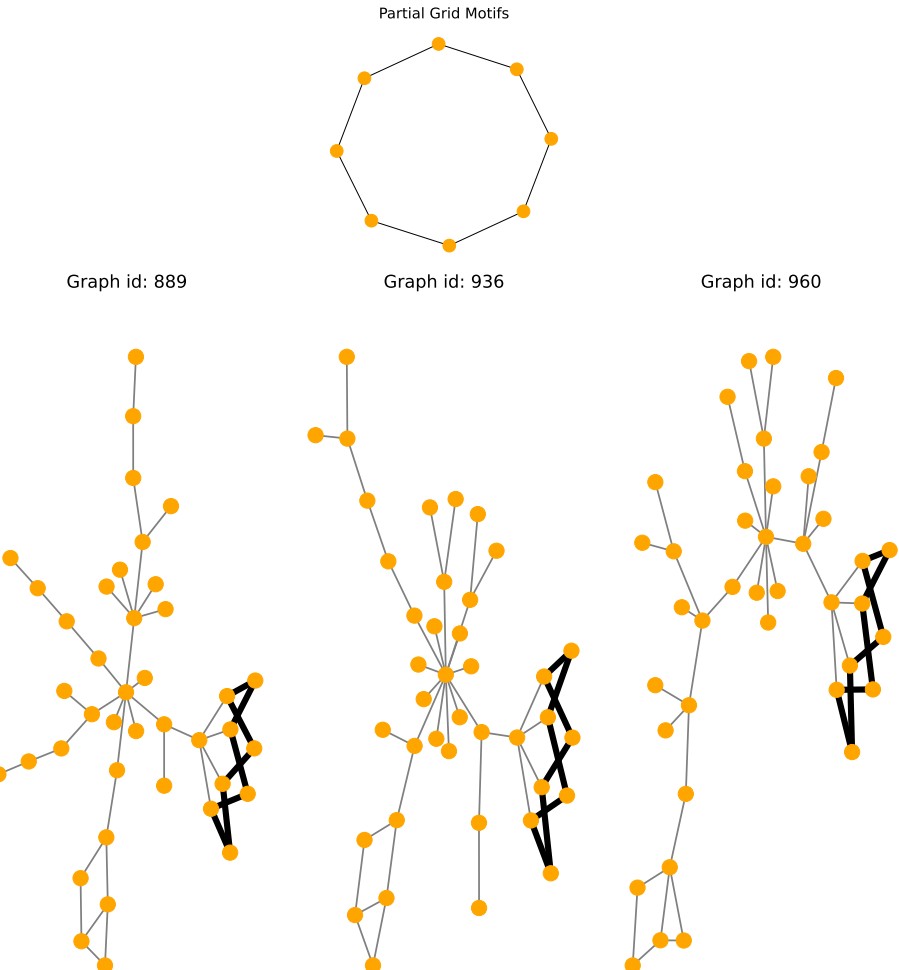

**Figure 9:** Illustration of the grid's external border only local explanation extracted for some samples of Class$_1$, along with three examples of instance-level local explanations with the selected explanation in bold. The expected local explanation is comprised by both the entire grid and the house.

Mutagenicity, instead, given the presence of informative node features, and given the absence of rich topological motifs, GAT and GCN perform comparably to GIN and SAGE.

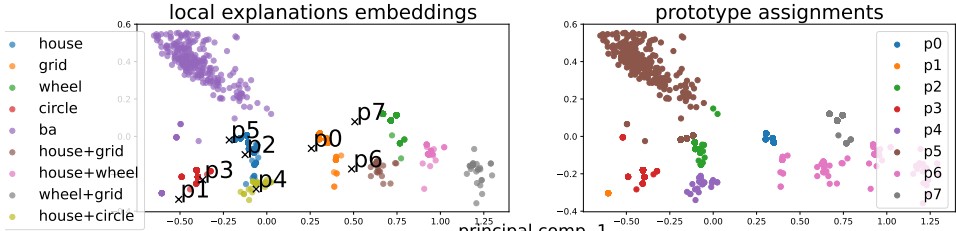

**Figure 10:** 2D view of the embedding space annotated with prototypes positions for the train split of BAMultiShapesMC.

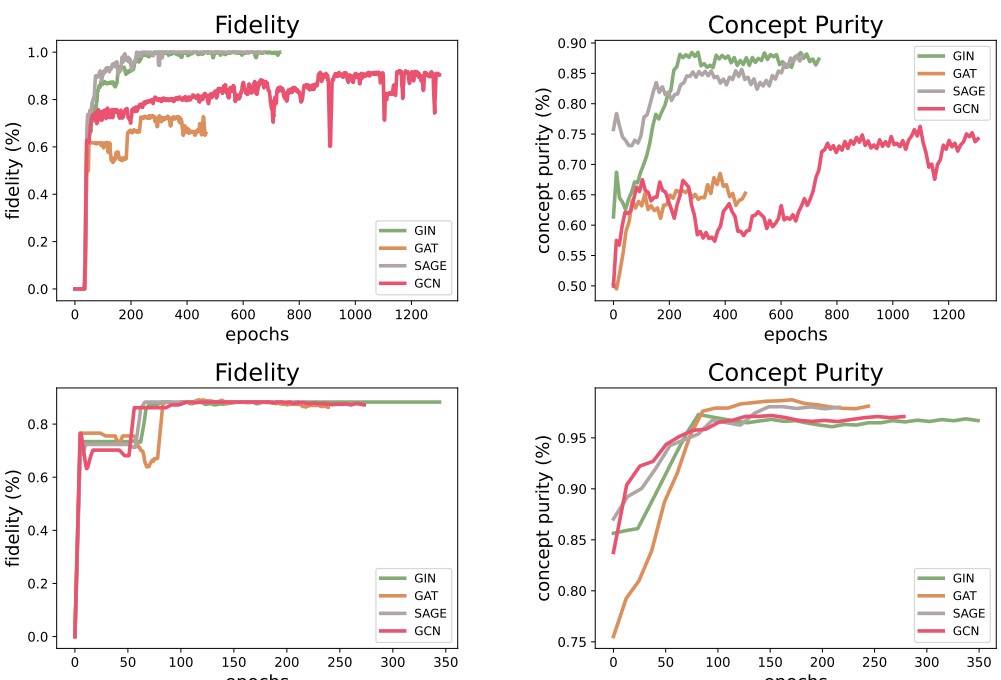

**Figure 11:** Ablation study on the architecture in use for the Local Explanations Embedder $h$. The first line is referred to BAMultiShapes, while the last to Mutagenicity

