# OpenReview forum: "Global Explainability of GNNs via Logic Combination of Learned Concepts"
_logconference.io/LOG/2022/Conference — LoG 2022 Poster_

### Official Review · Reviewer_gDWE · 2022-09-26

**Overall Score:** 8
**Confidence:** 3

**Review:**

In this work, the authors propose GLGExplainer (Global Logic-based GNN Explainer), capable of generating explanations as arbitrary Boolean combinations of learned graphical concepts. The contributions of this work can be summarized as follows: (i) The proposed method provides a Global Explanation in terms of logic formulas, extracted by combining in a fully differentiable manner graphical concepts derived from local explanations. It is flexible and can introduce various models that can provide subgraph interpretations. (ii) The proposed method is faithful to the data domain as it does not require prior knowledge. (iii) The authors conducted various experiments, and the results demonstrated that the proposed method produces highly interpretable explanations.

Strongest points:
1. The paper is well-written. The motivation is clear, as the authors clearly summarize previous achievements and limitations of GNN interpretability.
2. The proposed approach is novel. It consists of multiple steps that are clear, easy to understand and appear to be effective.
3. The authors conducted experiments on two datasets, and the experimental results well support the effectiveness of the proposed method in providing remarkable accuracy and highly interpretable explanations. To help better understand the proposed method, the authors also performed analyses of the Focal Loss, the number of prototypes, and the Discretization trick.

Suggestions:
1. Since few similar works are available to evaluate the effectiveness of the interpretability of the proposed model, it would be beneficial to provide more experiments. For example, In the BAMultiShapes dataset, the authors may consider a multi-class classification problem that involves various combinations of motifs.
2. The authors adopt the GCN model and train GLGExplainers to explain the predictions made by the GCN. From the results shown in Table 3 and Figure 2, we can observe that the GCN failing at classifying some samples leads to the GLGExplainer not being able to discover some structures as explanations. It would be helpful to consider other GNN models that produce higher accuracy. In this case, the authors can test more comprehensively the ability of GLGExplainer to capture the interpretation of various combinations of concepts.

I would like to recommend accepting this paper. This paper presents a novel idea and an effective explainable method that has the potential to be studied in many domains. Ideally, the proposed method can be applied to explain any GNN that has been trained. The experiments are thorough and help the readers fully understand the characteristics of the model.

There is a typo in line 127: intepretable—> interpretable.

---

### Official Review · Reviewer_rzN4 · 2022-10-11

**Overall Score:** 6
**Confidence:** 4

**Review:**

**Summary**

The paper presents a novel method for generating global explanations for Graph Neural Networks (GNNs). There have been a plethora of local explanation methods that generate instance-level explanations for GNN model predictions. However, there has been little to no work done on generating a global explanation for the behavior of a GNN. The authors propose Global Logic-based GNN Explainer (GLGExplainer), the first Global Explainer capable of generating global explanations by combining learned graph concept vectors. In particular, GLGExplainer uses individual local explanations as inputs and combines them using boolean operations to learn concepts (group of local explanations). Results on two datasets show the effectiveness of the proposed approach over existing methods.

**Strengths**

1. The paper presents a new approach to generating global explanations for GNNs using a cluster of local explanations.
2. The generated explanations provide logic formulas expressed over learned graphical concepts, which provide concise and faithful explanations.

**Weakness and Open Questions**
1. It would be great to describe the evaluation metrics in detail. In particular, i) given that GLGExplainer is optimized on the original predictions, is it efficient to use fidelity as an evaluation metric because we can expect a higher score if the loss objective converges, and ii) the definition of concept purity metric is unclear?
2. It is unclear whether the concepts for different categories are unique. For instance, Figure 3 shows a high overlap between explanations from Class 0 and Class 1.
3. The GLGExplainer framework needs more description in terms of the individual components and their respective motivation. Formally, we first generate local explanations for the decisions of the original GNN model $f$ using any subgraph-based local explanation method. Then, we generate the embeddings of these individual local explanations and transform them into a concept vector using another GNN model $h$. Hence, GLGExplainer is approximating the behavior of the underlying complex GNN model $f$ using model $h$. Given that $h$ itself is a complex multi-layer GIN network (in experiments), i) how reliable are the final global explanations? and ii) does the final explanation depend on the choice of the GNN model $h$?
4. Do the GLGExplainer explanations depend significantly on the quality of the local explanations, or is it agnostic to the choice of local GNN explainers?

---

### Official Review · Reviewer_GYBy · 2022-10-21

**Overall Score:** 5
**Confidence:** 5

**Review:**

In this paper, the authors propose GLGExplainer to investigate the global explainability of GNN in the form of logical combinations. GLGExplainer generates boolean combinations of local explanations over graphical concepts as the global explanation. Experiments are evaluated on both synthetic and real-world datasets.

**Strengths:**
1. It is interesting that this paper uses boolean combinations as a global explanation of GNN, while most existing methods generate a local explanation or a synthetic prototypical graph as a global explanation.
2. The paper is written clearly.

**Weaknesses:**

1. In the Introduction, it is said that "Global Explanation problem for GNN as a form of input optimization", while the citation paper [1] is not related to GNN, which is confusing.
2. The motivation of this paper is not clear, and we would like to see more concrete scenarios to clarify the need for a model-level explanation of GNN.
3. More related works on instance-level explanation need to be introduced, such as GraphLIME [2], GEM [3], and OrphicX [4].
4. We believe the idea of formulating explanations is interesting. However, it is not novel that has been used in GLocalX, and this work just combines the explanations from PGExplainer, which already has a good performance.
5. Fidelity is a widely used metric to evaluate the performance of a GNN explainer, but in this paper, there is no detailed description of it. In Table 1, it is shown that fidelity and formula accuracy have the same value. A clear statement of "formula accuracy" and "concept purity" should be provided.
6. The class probability of the explanation graphs predicted by the GNN is also an important metric used in XGNN, and it would be more intuitive if the authors applied this metric and compared the results with XGNN in experiments.
7. We'd like to see experimental results on more datasets, especially multi-class datasets.

[1] Towards Global Explanations of Convolutional Neural Networks with Concept Attribution. CVPR, 2020.
[2] GraphLIME: Local Interpretable Model Explanations for Graph Neural Networks. IEEE TKDE, 2022.
[3] Generative Causal Explanations for Graph Neural Networks. ICML, 2021.
[4] OrphicX: A Causality-Inspired Latent Variable Model for Interpreting Graph Neural Networks. CVPR, 2022.

**Minor points:**
1. In figure 4, the legend is wrong. The orange line should be "W/o Focal loss".
2. Figure 1 illustrates the proposed method, but there is too much content in the figure, and the captain is not detailed enough to describe the model. Also, more illustrations are required for the figures, such as figure 4, figure 5, and figure6.

---

### Official Review · Reviewer_zxei · 2022-10-22

**Overall Score:** 8
**Confidence:** 4

**Review:**

This work addresses the global explainability of GNNs, proposing a method that allows deriving class-based explanations in contrast to the more common instance-based methods.

The idea is novel and very interesting. It proposes to embed local explanations generated through local methods, map them in "concepts", and then map concepts to classes through a logic explainable network.

Strengths:
- The method and the experiments are explained well, also through the extensive appendix, providing key details.
- The considered metrics allow capturing different aspects of the learned explanations.

Weaknesses:
- The evaluation includes only toy datasets with a limited number of classes/motifs. In particular, all the classes can be characterized by the presence/absence of a specific motif (or a combination of up to 3). It would be interesting to study complex classes (that could be automatically generated, and precisely evaluated), and their impact on the number of concepts needed. Or, the presence of more heterogeneous motifs (subgraphs with very different sizes). Currently, the included experiments do not allow concluding that this method would work for real-world datasets and applications.

---

### Meta-Review · Area_Chair_sdcQ · 2022-11-19

**Confidence:** 5
**Recommendation:** Accept

**Meta Review:**

This paper studies global explainability of GNNs, and proposes a new approach involving local method, concept generation, and logical combination. The paper is novel and well written. The reviewers were originally concerned about experiments, including metrics and graph tasks. The authors have responded to those feedback with new experiments and updated writing. The reviewers unanimously vote for accept.

---

### Decision · Program_Chairs · 2022-11-23

Accept (Poster)